

# Unbiased high-throughput characterization of mussel transcriptomic responses to sublethal concentrations of the biotoxin okadaic acid

Victoria Suarez-Ulloa[1], Juan Fernandez-Tajes[2], Vanessa Aguiar-Pulido[3], M. Veronica Prego-Faraldo[1,4], Fernanda Florez-Barros[5], Alexia Sexto-Iglesias[4], Josefina Mendez[4] and Jose M. Eirin-Lopez[1]

[1] Chromatin Structure and Evolution Group (Chromevol), Department of Biological Sciences, Florida International University, Miami, FL, United States of America
[2] McCarthy Group, Wellcome Trust Center for Human Genetics, University of Oxford, Oxford, United Kingdom
[3] Bioinformatics Research Group (BioRG), School of Computing & Information Sciences, Florida International University, Miami, FL, United States of America
[4] Xenomar Group, Department of Cellular and Molecular Biology, University of A Coruña, A Coruña, Spain
[5] Centre for Nephrology, Royal Free Hospital, University College London, London, United Kingdom

Corresponding author
Jose M. Eirin-Lopez,
jeirinlo@fiu.edu

## ABSTRACT

**Background.** Harmful Algal Blooms (HABs) responsible for Diarrhetic Shellfish Poisoning (DSP) represent a major threat for human consumers of shellfish. The biotoxin Okadaic Acid (OA), a well-known phosphatase inhibitor and tumor promoter, is the primary cause of acute DSP intoxications. Although several studies have described the molecular effects of high OA concentrations on sentinel organisms (e.g., bivalve molluscs), the effect of prolonged exposures to low (sublethal) OA concentrations is still unknown. In order to fill this gap, this work combines Next-Generation sequencing and custom-made microarray technologies to develop an unbiased characterization of the transcriptomic response of mussels during early stages of a DSP bloom.

**Methods.** Mussel specimens were exposed to a HAB episode simulating an early stage DSP bloom (200 cells/L of the dinoflagellate *Prorocentrum lima* for 24 h). The unbiased characterization of the transcriptomic responses triggered by OA was carried out using two complementary methods of cDNA library preparation: normalized and Suppression Subtractive Hybridization (SSH). Libraries were sequenced and read datasets were mapped to Gene Ontology and KEGG databases. A custom-made oligonucleotide microarray was developed based on these data, completing the expression analysis of digestive gland and gill tissues.

**Results.** Our findings show that exposure to sublethal concentrations of OA is enough to induce gene expression modifications in the mussel *Mytilus*. Transcriptomic analyses revealed an increase in proteasomal activity, molecular transport, cell cycle regulation, energy production and immune activity in mussels. Oppositely, a number of transcripts hypothesized to be responsive to OA (notably the Serine/Threonine phosphatases PP1 and PP2A) failed to show substantial

modifications. Both digestive gland and gill tissues responded similarly to OA, although expression modifications were more dramatic in the former, supporting the choice of this tissue for future biomonitoring studies.

**Discussion.** Exposure to OA concentrations within legal limits for safe consumption of shellfish is enough to disrupt important cellular processes in mussels, eliciting sharp transcriptional changes as a result. By combining the study of cDNA libraries and a custom-made OA-specific microarray, our work provides a comprehensive characterization of the OA-specific transcriptome, improving the accuracy of the analysis of expresion profiles compared to single-replicated RNA-seq methods. The combination of our data with related studies helps understanding the molecular mechanisms underlying molecular responses to DSP episodes in marine organisms, providing useful information to develop a new generation of tools for the monitoring of OA pollution.

## INTRODUCTION

Harmful Algal Blooms (HABs) constitute an environmental phenomenon encompassing critical relevance due to their increasing frequency and impact in coastal areas (*Anderson, 2009*). Diarrhetic Shellfish Poisoning (DSP) blooms represent a major threat in widespread geographic areas comprising the Atlantic coast of Europe, Chile and Japan (*Reguera et al., 2014*), where natural outbreaks of toxic *Dinophysis* and *Prorocentrum* microalgae produce large amounts of DinophysisToXins (DTXs) and Okadaic Acid (OA) biotoxins (*Sellner, Doucette & Kirkpatrick, 2003*). OA is the primary cause of acute DSP intoxication of human consumers of shellfish, causing strong economic losses for the aquaculture industry. This biotoxin constitutes a well-known phosphatase inhibitor encompassing tumorigenic and apoptotic effects, even at low concentrations (*Prego-Faraldo et al., 2015*). Indeed, OA is capable of inducing genotoxic and cytotoxic damage, representing a hazard under chronic exposure conditions (*Prego-Faraldo et al., 2013*; *Valdiglesias et al., 2013*).

Given the noted risks of OA for human health and marine ecosystems, DSP events represent one of the most important threats for the shellfish aquaculture industry. Consequently, important efforts have been dedicated to develop rapid and sensible DSP biomonitoring methods, most notably using bivalve molluscs (e.g., mussels, oysters, clams, etc.) as sentinel organisms (*Manfrin et al., 2010*; *Fernandez-Tajes et al., 2011*; *McNabb et al., 2012*; *Romero-Geraldo, Garcia-Lagunas & Hernandez-Saavedra, 2014*; *Huang et al., 2015*). The choice of these organisms is supported by their wide distribution, sessile and filter-feeding lifestyles as well as their ability to accumulate high amounts of biotoxins, while displaying a particularly strong resilience to their harmful effects (*Svensson, Sarngren & Forlin, 2003*; *Prado-Alvarez et al., 2012*; *Prado-Alvarez et al., 2013*). During

the last decade, the increasing availability of genomic resources in bivalves has improved classical biomonitoring approaches (e.g., quantification of biotoxin content in mollusc tissues), notably by developing molecular high-throughput studies evaluating omic (transcriptomic and proteomic) responses to HAB stress and their potential biomarker application (*Manfrin et al., 2010*; *Suarez-Ulloa et al., 2013a*; *Gerdol et al., 2014*; *Huang et al., 2015*). Nonetheless, while this approach has proven to be a promising venue for pollution biomonitoring (*Campos et al., 2012*; *Suarez-Ulloa et al., 2013b*), additional efforts are still required to clarify the cause–effect relationships between environmental stressors and changes in gene expression patterns. In doing so, it will be possible to transform the extraordinary amount of molecular data resulting from omic experiments into a practical tool for marine pollution biomonitoring.

Mussels start accumulating OA in their tissues during early stages of DSP blooms, however, their commercialization is still allowed by the applicable legislation as long as the concentration of this biotoxin does not exceed the legal threshold of 160 μg OA equivalents/kg shellfish meat (European Union legislation). Nonetheless, it has been demonstrated that exposure to low OA concentrations for short periods of time is enough to produce genotoxic and cytotoxic effects *in vitro* (*Prego-Faraldo et al., 2015*). The present work aims to provide a better understanding of the molecular mechanisms underlying the environmental responses of bivalve molluscs to sublethal concentrations of OA. For this purpose, Next-Generation sequencing and custom-made microarray technologies were combined to develop an unbiased characterization of the transcriptomic response of bivalve molluscs (mussels) to OA during early stages of a DSP bloom. These analyses build on previous studies (including our own) focused on specific subsets of genes (i.e., chromatin structure/function (*Suarez-Ulloa et al., 2013a*); oxidative stress, cell cycle regulation and immune response (*Romero-Geraldo, Garcia-Lagunas & Hernandez-Saavedra, 2014*; *Romero-Geraldo & Hernandez-Saavedra, 2014*)), as well as on the application of microarray technology to study the OA-specific transcriptome (*Manfrin et al., 2010*). Our results expand the scope, dimension and methodological approaches of these studies, improving the description of the cellular processes involved in the mussel response to OA toxicity. In doing so, this study generates omic information useful for identifying molecular signatures of marine pollution during DSP blooms. Contrary to quantitative analytical methods (i.e., LC-MS), this approach selectively identifies stressors of very different nature, assessing the magnitude of the toxic effects for organisms and communities. In addition, it provides further insights into the molecular strategies underlying the extraordinary resilience of bivalve molluscs to environmental stress.

## METHODS

### Specimen collection and experimental simulation of DSP HABs

Mussel specimens (*Mytilus galloprovincialis* (Lam.)) were collected in Valcobo beach, Galicia, NW Spain (43°19′02.71″N 8°21′56.35″W) in an area free of OA pollution during the resting period of the reproductive cycle (*Banni et al., 2011*). Sampled individuals (adults between 10 and 15 cm) were randomly divided into two experimental groups;

exposed (exposed group) and non-exposed (control group) to the OA-producing dinoflagellate *Prorocentrum lima.* Both groups were kept in aerated seawater tanks and fed continuously with a suspension of the microalgae *Tetraselmis suecica* and *Isochrysis galbana.* After acclimation (one week), the exposed group was fed with 200 cells/L of a *Prorocentrum lima* culture (exponential phase) for 24 h. Specimens were dissected immediately after exposure, collecting samples from digestive gland and gill tissues. Each experimental sample consisted of tissue obtained from 5 individuals per group, dissected and pooled for RNA extraction.

## RNA extraction and construction of cDNA libraries

The OA content in exposed and control samples was quantified using high-resolution mass spectrometry (*Domenech et al., 2014*). Total RNA was extracted from digestive gland and gill tissues using TRIzol® (Thermo Scientific, Waltham, Massachusetts, USA) following the manufacturer's instructions. RNA concentration and quality check was measured using a NanoDrop spectrophotometer (Thermo Scientific, Waltham, Massachusetts, USA) and a Bioanalyzer (Agilent Technologies, Santa Clara, California, USA). cDNA library construction and pyrosequencing were performed using digestive gland samples, based on the larger absorption and accumulation of OA in this tissue. cDNA libraries were obtained from digestive gland tissue using the SMARTer™ PCR cDNA synthesis kit (Clontech, Mountain View, California, USA) and purified with GeneJET™ PCR Purification Kit (Thermo Scientific, Waltham, Massachusetts, USA) according to the manufacturer's instructions.

The construction of normalized cDNA libraries (norm), for both exposed (mgt) and control (mgc) samples was carried out using the Trimer cDNA Normalization Kit (Evrogen, Moscow, Russia) following manufacturer's protocol. This method enhances the detection of rare (lower concentration) transcripts by decreasing the prevalence of highly abundant transcripts (*Bogdanova et al., 2011*). The Suppression Subtractive Hybridization (SSH) libraries were constructed using the PCR-Select™ cDNA subtraction kit (Clontech, Mountain View, California, USA), following manufacturer's instructions. Accordingly, two types of SSH libraries were produced: forward (fwd) and reverse (rev), representing upregulated and downregulated transcripts, respectively. This method was used to optimize the isolation of differentially expressed transcripts by removing commonly abundant cDNAs (*Diatchenko et al., 1996*).

## Library sequencing and characterization

Normalized (exposed and control) and SSH (forward and reverse) libraries were sequenced by means of Roche-454 FLX+ Titanium pyrosequencing (Roche Diagnostics, Indianapolis, Indiana, USA), with a sequencing depth of 40×. The obtained read datasets were preprocessed, assembled *de novo* and mapped to Gene Ontology (GO) and KEGG databases (*Kanehisa, 2002*). Additionally, low quality reads were discarded, and adaptors and low quality ends were trimmed before *de novo* assembly using MIRA v.3.9.16 (*Chevreux, Wetter & Suhai Suhai, 1999*). Both normalized and SSH read datasets are available at NCBI's Bioproject database under the accession number PRJNA167773.

The generated contigs were annotated using BLAST (blastx) against the non-redundant protein sequence database (nr), setting a threshold e-value of $1e^{-6}$ (*Altschul et al., 1997*). Contigs were subsequently annotated with GO terms using the Blast2GO suite (*Conesa et al., 2005*; *Gotz et al., 2008*), including those terms obtained from InterPro and Annex analyses (*Apweiler et al., 2001*; *Myhre et al., 2006*).

## Custom-made microarray construction and differential expression analysis

The sequencing and assembly of normalized and SSH libraries allowed to design specific probes targeting many of the transcripts identified. Accordingly, an Agilent oligonucleotide microarray encompassing 51,300 probes was constructed using the eArray™ design tool (Agilent Technologies, Santa Clara, California, USA) following a two-color Microarray-Based Gene Expression Analysis v.6.5 Agilent-specific protocol with dye swap. Two biological replicates per tissue sample were analyzed in microarray experiments. Expression analyses were conducted using the R package limma from the Bioconductor repository (*Ritchie et al., 2015*). Results are organized based on the magnitude of the observed change in expression or Fold Change in a logarithmic scale (logFC) and the statistical significance of the observed change in expression represented by an adjusted $p$-value or False Discovery Rate by the Benjamini–Hochberg procedure (FDR). Probes showing an FDR < 0.05 were considered as differentially expressed. The correlation between logFC values of differentially expressed transcripts commonly observed in both digestive gland and gill tissues was analyzed using a linear regression based on Pearson's coefficient of determination. The GO terms for the most representative biological processes in both upregulated and downregulated groups of transcripts were determined using topGO with statistical significance ($p$-values) calculated according to the weight algorithm (*Alexa & Rahnenfuhrer, 2010*). Lastly, contigs were also mapped to the KEGG database for pathway analysis (*Kanehisa, 2002*).

# RESULTS AND DISCUSSION

## Characterization of OA-specific cDNA libraries in the mussel *Mytilus*

The analysis of OA in pooled digestive gland tissue of exposed individuals revealed a concentration of 18.27 ng of OA per gram of fresh tissue in exposed individuals (OA content in controls individuals is below detection limit), an order of magnitude below the legal OA limit established for safe consumption of shellfish in the European Union (*Reguera et al., 2014*). This result reinforces the focus of the present study on early stages of DSP HAB episodes, at a moment when mussels start accumulating OA in their tissues but their commercialization is still allowed by law. The construction of normalized (norm) cDNA libraries yielded 919,177 good quality reads, 514,276 for the exposed group (mgt) and 404,901 for the control group (mgc). After assembly, a total of 24,624 and 16,395 consensus sequences (contigs) were obtained, respectively. Complementary, the SSH libraries produced a set of 1,221,928 good quality reads (SSH) with 469,795 corresponding to the forward (fwd) library and 752,133 to the reverse (rev) library. Once assembled,

**Table 1 Reads and annotated contigs obtained from the cDNA libraries constructed.**

| | Normalized libraries | | SSH libraries | |
| --- | --- | --- | --- | --- |
| | Exposed | Control | Forward | Reverse |
| Reads | 514,276 | 404,901 | 469,795 | 752,133 |
| Contigs | 24,624 | 16,395 | 21,591 | 33,437 |
| Annotated contigs | 10,617 (43%) | 7,335 (45%) | 6,448 (30%) | 18,553 (55%) |

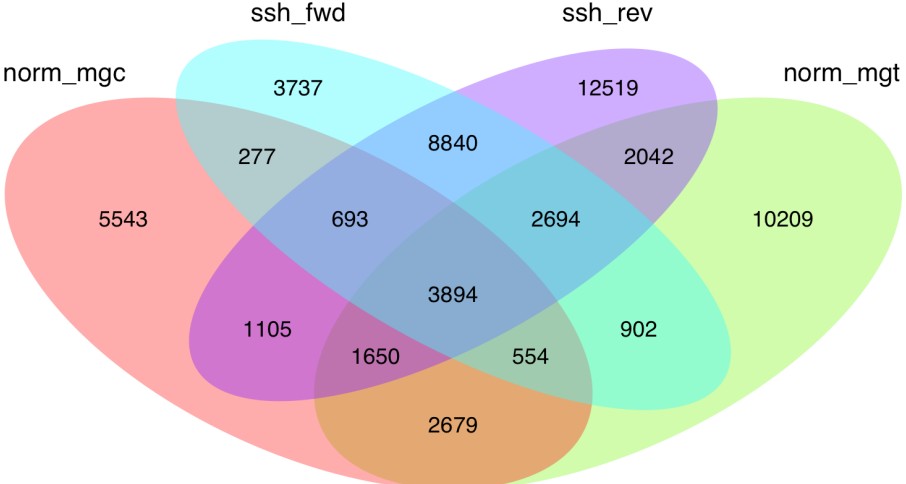

**Figure 1 Venn diagram showing the extent of redundancy between the different libraries constructed in the present work: norm_mgc, normalized control library; norm_mgt, normalized exposed library; ssh_fwd, SSH forward library; ssh_rev, SSH reverse library.**

a total of 21,591 contigs and 33,437 contigs were obtained, respectively. Overall, blastx searches against the nr database resulted in the identification of 17,952 contigs from normalized libraries and 25,001 contigs from SSH libraries (see details in Table 1).

Given the high level of redundancy among *de novo* assembled libraries (Fig. 1), contigs were combined into unigenes according to their annotation and were considered equivalent to the annotated transcripts (unigenes are therefore considered a set of uniquely identified transcripts). The normalized and SSH libraries constructed expand and complement partial sequence data previously released by our group in the Chromevaloa database (*Suarez-Ulloa et al., 2013a*). By combining both sets of sequences, the present work was able to produce a microarray tool increasing the coverage of OA-specific transcriptome in the mussel *Mytilus*, improving the unbiased analysis of the differences in gene expression.

## Microarray-based analysis of transcriptomic responses to OA

The present work expanded previous analysis of the mussel's response to OA exposure using an omic approach using an oligonucleotide microarray designed from the sequences identified in pyrosequencing libraries. Accordingly, a medium-high coverage Agilent

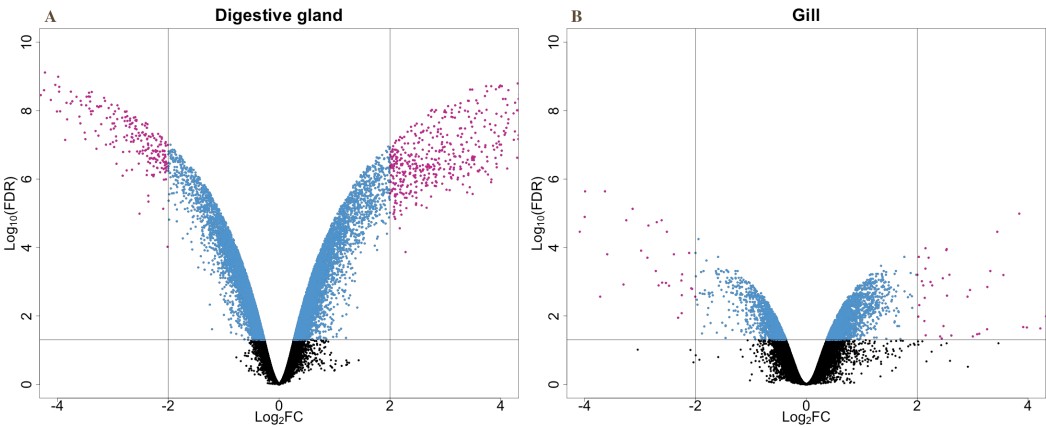

**Figure 2** **V-plots showing gene expression differences detected through microarray analysis in digestive gland (A) and gill (B) tissues.** These differences are represented as net expression change (logFC) with statistical significance (FDR) indicated as a logarithmic scale. Probes highlighted in blue (FDR < 0.05) and purple (FDR < 0.05 and logFC > 2) represent the groups of transcripts displaying largest changes in gene expression between exposed and control treatments.

microarray (51,300 probes) was designed and developed using the sequences (contigs) obtained from the cDNA libraries constructed in this work. The hybridization of the microarray with RNA samples from exposed and control groups revealed a total number of 14,160 probes (digestive gland) and 6,913 probes (gill) differentially expressed (Fig. 2). The consistency between expression profiles in digestive gland and gill was assessed performing a linear regression of the logFC values of differentially expressed transcripts common for both tissues (i.e., those showing FDR < 0.05 in both cases), showing a good correlation between both sets of transcripts (Fig. 3). The detailed description of the transcripts displaying the highests differences in expression levels in both tissues, along with the maximum observed logFC value in the microarray analysis, is indicated in Supplemental Information 1 and 2.

The microarray analysis identified a set of transcripts displaying sharp expression differences between exposed and control treatments Supplemental Information 1 and 2, expanding the list of transcripts potentially involved in the response to OA (*Manfrin et al., 2010*; *Suarez-Ulloa et al., 2013a*). This was primarily facilitated by a larger coverage in the transcriptomic assessment, but also by the increase in bivalve genomic information that has been incorporated to molecular databases in recent years (*Suarez-Ulloa et al., 2013b*; *Gerdol et al., 2014*). Differentially expressed transcripts identified in this study include heat shock 70 kda protein 12b, proteases like cathepsins b and d, polyubiquitin and and proteasomal subunit beta type-4, commonly associated with an accumulation of misfolded or oxidized proteins observed under different types of environmental stress (*Gotze et al., 2014*). A subset of the identified transcripts showing the highest fold-change classified according to their main functional role is presented in Table 2. Our results corroborate previous analyses describing the responses of *Mytilus galloprovincialis* to OA stress (*Manfrin et al., 2010*), particularly the strong upregulation of vdg3 and elongation factor 2. In the case of vdg3, this gene is associated with developmental changes during

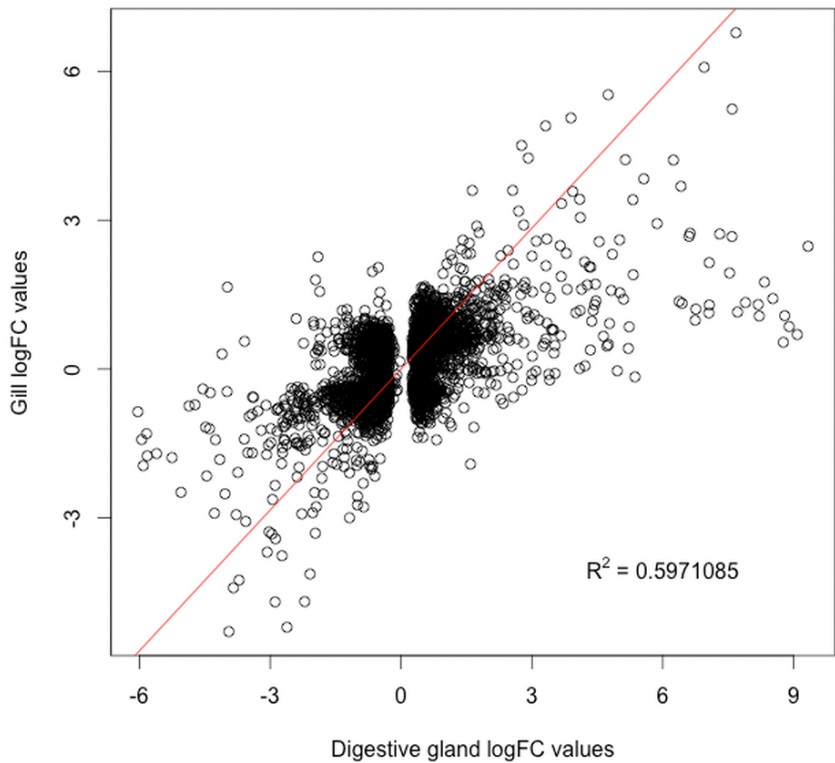

**Figure 3 Correlation between paired logFC values calculated for transcripts identified in digestive gland and gill tissues between exposed and control treatments.** Overall, a good level of agreement is found for gene expression changes ($R^2 \cong 0.6$).

the benthic settlement stage (*He et al., 2014*) and it has only been identified in bivalves, being particularly abundant in the digestive gland. On the other hand, the elongation factor 2 (EEF-2) is widely ubiquitous across eukaryotic taxa, playing an essential regulatory role in protein synthesis as a housekeeping gene. Although the functional implications of vdg3 and EEF-2 in the context of this study are still unclear, the obtained results support previous reports discouraging the use of EEF-2 as an internal control for qPCR analyses on bivalves without further validation (*Du et al., 2013*).

Opposite to these findings, a number of transcripts hypothesized to be responsive to OA failed to show substantial expression modifications under the conditions of this study. Notably, the Serine/Threonine phosphatases PP1 and PP2A, specific targets in OA toxicity mechanisms, did not show significant expression changes between treatments. OA is a well known selective inhibitor of the enzymatic activity of PP1 and PP2A phosphatases with critical consequences for the cell's fate (*Shenolikar, 1994*). However, our results suggest that the upregulation of the PP1 and PP2A genes is not a relevant strategy versus the antagonist effects of OA. Similarly, Multi-Xenobiotic Resistance proteins (MXRs), good candidates to explain the high tolerance of bivalves versus pollution (*Contardo-Jara, Pflugmacher & Wiegand, 2008*), failed to show significant changes in expression. It is possible that their attributed role in OA uptake could be supplied by other proteins (e.g., the highly upregulated nose resistant to fluoxetine protein 6, a transport mediator of xenobiotics

**Table 2** Selected subsets of differentially expressed transcripts identified by microarray analysis representative of the following functional categories: (a) protein repair or degradation, (b) immune response, (c) transport and energy production and (d) cell cycle regulation.

| Protein repair/degradation | Immune response |
| --- | --- |
| Heat shock 70 kda protein 12b | Mytimacin-5 |
| Cathepsin d | c1q domain-containing protein 1q3 |
| Cathepsin b | c1q domain-containing protein 1q25 |
| Proteasome subunit beta type-4 | Mytimicin precursor |
| **Transport/energy production** | **Cell cycle regulation** |
| Nose resistant to fluoxetine protein 6 | bcl2 adenovirus e1b 19-kd protein-interacting |
| Interferon-inducible GTPase 5-like | Apoptosis inhibitor iap |
| nadh dehydrogenase subunit | jagged 1 |
| Atpase H+ transporting lysosomal 21 kda v0 subunit | Oncoprotein-induced transcript 3 protein |

accross tissues). Indeed, lysosomal uptake has been suggested as a possible explanation for the extraordinary tolerance of mussels to the effects of DSP pollution (*Svensson, Sarngren & Forlin, 2003*).

In addition to transcripts previousy linked to OA responses, our results found an upregulation of an antimicrobial peptide (mytimacin) as well as an antifungal peptide (mytimycin) specific from mussels (*Sonthi et al., 2011*; *Gerdol et al., 2012*). Interestingly, mytimacin-5 (partial) was identified as one of the most upregulated transcripts in both gill and digestive gland. This peptide is especially interesting among the mytimacin family due to two additional cysteines in conserved positions predicted to form an extra disulfide bridge with yet unknown functional implications (*Gerdol et al., 2012*). C1q domain-containing proteins 1q3 and 1q25 showed a strong upregulation in the digestive gland. C1q is involved in the mammalian classical component pathway, playing an important role in innate immunity. Although no clear homologues to the vertebrate C1q complex subunits have been found in invertebrates yet, a massive expansion in the C1q domain-containing protein family has been suggested in bivalves, including *Mytilus* (*Gerdol, Venier & Pallavicini, 2015*). C1q domain-containing proteins are very versatile and might display a wide range of ligand interactions and functions such as clearance of apoptotic cells through direct binding (*Kishore et al., 2004*). They have been found upregulated in molluscs challenged with different pathogens (*Perrigault, Tanguy & Allam, 2009*; *Taris et al., 2009*). Although their specific function remains unclear, the substantial upregulation found in the present work might be indicative of a relevant role during environmental stress responses.

Altogether, the obtained results provide valuable insights into the molecular effects of OA in the mussel *Mytilus* and will be improved by considering the following: (a) kinetic effects during transcription and its regulation could bias estimations of differential expression (*Bai et al., 2015*); (b) the strong upregulation observed in endo-beta xylanases and endo-beta glucanases (although coherent with energy production) might be emphasized by the composition of the cell wall from dinoflagellates; (c) the differential

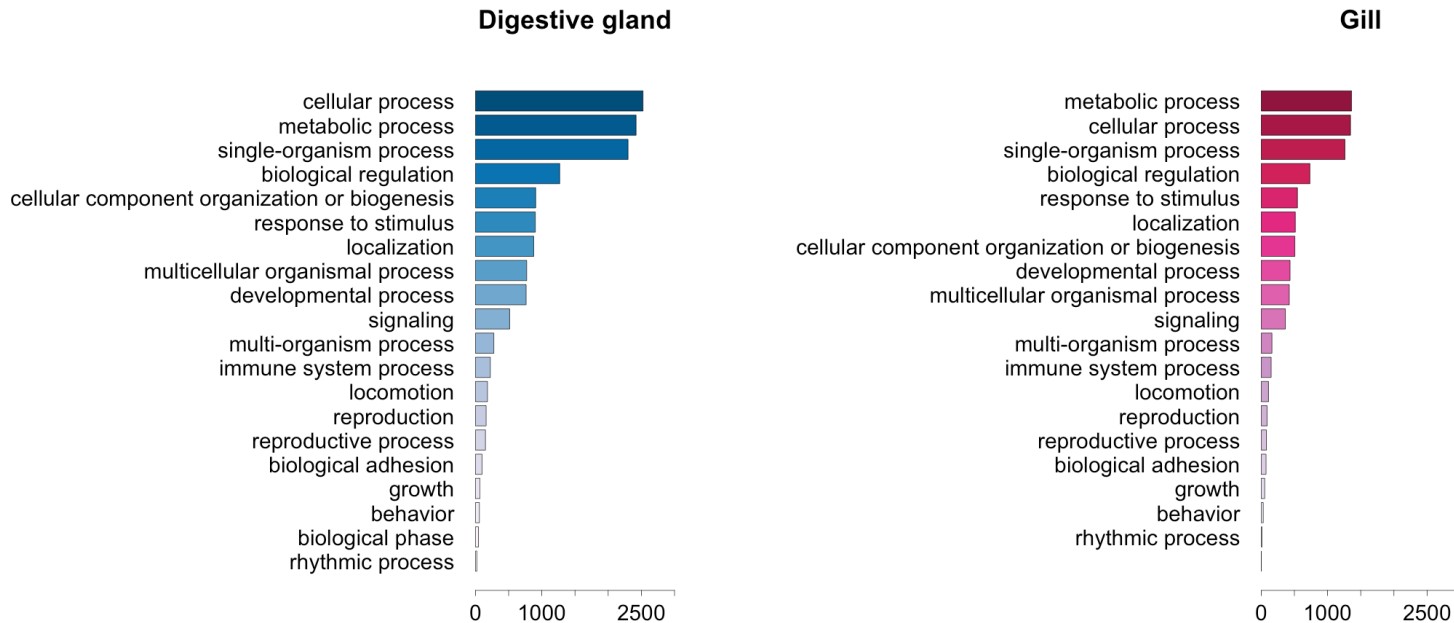

**Figure 4 Graphical representation of the GO terms (general sub-categories in Biological Process ontology) most represented in transcripts differentially expressed for each mussel tissue according to the microarray analysis.** The length of the bars is proportional to the number of sequences annotated for each specific GO term.

regulation of antimicrobial and antifungal peptides might be influenced by the presence of infiltrated hemocytes in digestive gland tissue.

## Expression and function profiles of transcripts differentially expressed in response to OA

The GO term annotation of transcripts differentially expressed in response to OA allowed the analysis of the biological processes in which their enconding genes are involved. A comparison of the functional profile for the two tissues studied is shown in Fig. 4. These profiles are based on the levels of representation for the most general sub-categories in GO stemming from Biological Process (*Ashburner et al., 2000*). Although absolute differences in magnitude are evident between digestive gland and gill (Fig. 2), no major functional differences were found when comparing the profiles for both tissues (Figs. 3 and 4). Nonetheless, such comparison might be hampered by sample size differences (e.g., subtle tissue-specific differences could remain undetected) and the fact that the microarray could lack gill-specific transcripts. Indeed, recent reports suggest that OA might display tissue-specific effects. Accordingly, different cytotoxic effects of OA specific for different human cell types had been demonstrated *in vitro* (*Rubiolo et al., 2011*). Furthermore, it has been reported that mussel gills display higher sensitivity to OA than hemocytes after one hour exposure (*Prego-Faraldo et al., 2015*). Tissue specificity is further evidenced by comparisons among enriched GO terms determined for transcripts upregulated and downregulated in digestive gland and gill (Table 3).

GO terms related with transcription regulation and cell cycle are enriched in the set of transcripts downregulated in the digestive gland (e.g., transcription from RNA

**Table 3** Enriched GO terms in sets of differentially expressed transcripts in both digestive gland and gill tissues. Data is sorted based on *p*-value in increasing (*p*-values are calculated according to the weight algorithm in TopGO).

| GO term description | GO number | Annotated | Expected | *p*-value |
|---|---|---|---|---|
| **Digestive gland—upregulated** | | | | |
| Vesicle-mediated transport | GO:0016192 | 176 | 60.79 | 1.00E–09 |
| Maintenance of protein localization in endoplasmic reticulum | GO:0035437 | 16 | 5.53 | 3.80E–08 |
| Cellular response to glucose starvation | GO:0042149 | 19 | 6.56 | 5.70E–08 |
| Cellular modified amino acid metabolic process | GO:0006575 | 72 | 24.87 | 7.40E–08 |
| ER overload response | GO:0006983 | 15 | 5.18 | 1.10E–07 |
| Activation of signaling protein activity involved in unfolded protein response | GO:0006987 | 15 | 5.18 | 1.10E–07 |
| Cerebellum structural organization | GO:0021589 | 15 | 5.18 | 1.10E–07 |
| Regulation of cell migration | GO:0030334 | 47 | 16.23 | 2.40E–06 |
| Negative regulation of cellular response to growth factor stimulus | GO:0090288 | 23 | 7.94 | 2.80E–06 |
| Endosomal transport | GO:0016197 | 31 | 10.71 | 7.00E–06 |
| Pteridine-containing compound metabolic process | GO:0042558 | 17 | 5.87 | 7.00E–06 |
| Secretion by cell | GO:0032940 | 58 | 20.03 | 1.50E–05 |
| Coenzyme metabolic process | GO:0006732 | 117 | 40.41 | 1.70E–05 |
| Regulation of actin filament polymerization | GO:0030833 | 25 | 8.63 | 2.60E–05 |
| Cerebellar Purkinje cell layer development | GO:0021680 | 18 | 6.22 | 2.90E–05 |
| Cellular response to interleukin-4 | GO:0071353 | 27 | 9.33 | 3.00E–05 |
| Regulation of protein ubiquitination | GO:0031396 | 68 | 23.49 | 4.40E–05 |
| Negative regulation of protein polymerization | GO:0032272 | 15 | 5.18 | 4.60E–05 |
| Aminoglycan metabolic process | GO:0006022 | 29 | 10.02 | 0.00015 |
| Regulation of lipid metabolic process | GO:0019216 | 16 | 5.53 | 0.00017 |
| **Digestive gland—downregulated** | | | | |
| DNA metabolic process | GO:0006259 | 157 | 102.77 | 2.60E–09 |
| Ribonucleoprotein complex biogenesis | GO:0022613 | 130 | 85.1 | 4.20E–08 |
| mRNA processing | GO:0006397 | 70 | 45.82 | 1.80E–06 |
| Cilium morphogenesis | GO:0060271 | 53 | 34.69 | 4.10E–06 |
| Transcription from RNA polymerase II promoter | GO:0006366 | 186 | 121.76 | 4.90E–06 |
| Mitochondrial ATP synthesis coupled electron transport | GO:0042775 | 81 | 53.02 | 1.50E–05 |
| Mitotic nuclear division | GO:0007067 | 82 | 53.68 | 2.30E–05 |
| Inorganic cation transmembrane transport | GO:0098662 | 109 | 71.35 | 2.80E–05 |
| Chromosome organization | GO:0051276 | 167 | 109.32 | 7.20E–05 |
| Microtubule-based movement | GO:0007018 | 97 | 63.5 | 8.20E–05 |
| Cilium organization | GO:0044782 | 42 | 27.49 | 0.00018 |
| Positive regulation of ubiquitin-protein transferase activity | GO:0051443 | 32 | 20.95 | 0.00019 |
| Sodium ion transport | GO:0006814 | 46 | 30.11 | 0.00022 |
| Anaphase-promoting complex-dependent proteasomal ubiquitin-dependent protein catabolic process | GO:0031145 | 31 | 20.29 | 0.00028 |
| G1/S transition of mitotic cell cycle | GO:0000082 | 44 | 28.8 | 0.00029 |
| Mitotic S phase | GO:0000084 | 30 | 19.64 | 0.0004 |
| Chromatin remodeling | GO:0006338 | 39 | 25.53 | 0.0005 |

Table 3 (*continued*)

| GO term description | GO number | Annotated | Expected | *p*-value |
|---|---|---|---|---|
| Regulation of multi-organism process | GO:0043900 | 43 | 28.15 | 0.00058 |
| Cilium or flagellum-dependent cell motility | GO:0001539 | 17 | 11.13 | 0.00073 |
| Histone acetylation | GO:0016573 | 33 | 21.6 | 0.00078 |
| **Gill—upregulated** | | | | |
| Biological process | GO:0008150 | 926 | 516.95 | 1.60E–06 |
| Positive regulation of cell growth | GO:0030307 | 12 | 6.7 | 0.00087 |
| Carbohydrate metabolic process | GO:0005975 | 39 | 21.77 | 0.00149 |
| Cellular catabolic process | GO:0044248 | 30 | 16.75 | 0.04401 |
| Protein folding | GO:0006457 | 23 | 12.84 | 0.05753 |
| Protein polyubiquitination | GO:0000209 | 17 | 9.49 | 0.06629 |
| Lipid metabolic process | GO:0006629 | 24 | 13.4 | 0.09686 |
| Intracellular transport | GO:0046907 | 18 | 10.05 | 0.11899 |
| Nucleobase-containing compound catabolic process | GO:0034655 | 12 | 6.7 | 0.14577 |
| Proteolysis | GO:0006508 | 22 | 12.28 | 0.16789 |
| Single-organism developmental process | GO:0044767 | 94 | 52.48 | 0.19993 |
| Cellular macromolecular complex assembly | GO:0034622 | 11 | 6.14 | 0.20518 |
| Protein complex subunit organization | GO:0071822 | 11 | 6.14 | 0.20518 |
| Generation of neurons | GO:0048699 | 16 | 8.93 | 0.21463 |
| Vesicle-mediated transport | GO:0016192 | 21 | 11.72 | 0.21627 |
| Response to oxygen-containing compound | GO:1901700 | 10 | 5.58 | 0.28271 |
| Protein complex assembly | GO:0006461 | 10 | 5.58 | 0.31076 |
| Cell cycle | GO:0007049 | 17 | 9.49 | 0.31266 |
| Response to external stimulus | GO:0009605 | 17 | 9.49 | 0.31266 |
| Positive regulation of transcription, DNA-templated | GO:0045893 | 12 | 6.7 | 0.32401 |
| **Gill—downregulated** | | | | |
| Microtubule-based process | GO:0007017 | 25 | 7.31 | 1.40E–07 |
| Energy derivation by oxidation of organic compounds | GO:0015980 | 10 | 2.92 | 0.0012 |
| Heterocycle biosynthetic process | GO:0018130 | 35 | 10.23 | 0.0041 |
| Aromatic compound biosynthetic process | GO:0019438 | 35 | 10.23 | 0.0041 |
| Cellular nitrogen compound biosynthetic process | GO:0044271 | 37 | 10.82 | 0.0086 |
| Organic cyclic compound biosynthetic process | GO:1901362 | 38 | 11.11 | 0.0121 |
| Biological process | GO:0008150 | 926 | 270.74 | 0.0161 |
| Regulation of signal transduction | GO:0009966 | 18 | 5.26 | 0.0165 |
| Cellular protein modification process | GO:0006464 | 57 | 16.67 | 0.0537 |
| Nucleotide metabolic process | GO:0009117 | 11 | 3.22 | 0.0687 |
| Response to organic substance | GO:0010033 | 26 | 7.6 | 0.1048 |
| Single-organism transport | GO:0044765 | 66 | 19.3 | 0.1202 |
| Cell morphogenesis involved in differentiation | GO:0000904 | 10 | 2.92 | 0.1363 |
| Regulation of multicellular organismal process | GO:0051239 | 10 | 2.92 | 0.1363 |
| Purine-containing compound metabolic process | GO:0072521 | 10 | 2.92 | 0.1363 |
| Single-organism biosynthetic process | GO:0044711 | 19 | 5.56 | 0.1603 |
| Cell surface receptor signaling pathway | GO:0007166 | 31 | 9.06 | 0.1635 |
| Regulation of biological quality | GO:0065008 | 17 | 4.97 | 0.202 |
| Anatomical structure morphogenesis | GO:0009653 | 33 | 9.65 | 0.2438 |
| Protein modification by small protein conjugation | GO:0032446 | 28 | 8.19 | 0.2836 |

polymerase II promoter, histone acetylation, mitotic nuclear division, mitotic S phase). On the contrary, these terms are mostly represented in the upregulated set of transcripts in the gill (e.g., positive regulation of cell growth, cell cycle, positive regulation of transcription, DNA-templated). Although this might suggest a higher degree of stress in digestive gland, both tissues consistently show enrichment in GO terms connected to DNA repair and degradation of damaged proteins. Therefore, while the mechanisms involved in OA toxicity could be consistent across these tissues, different responses could be elicited depending on the level of the accumulation. Further research will be required to clarify the extent in which the effects of OA are determined by the nature of the tissue, the time/dose or a combination of both.

Our results show an overall larger number of upregulated transcripts compared with those downregulated, in agreement with previous reports although a strong dependence of the expresion profiles with time was demonstrated (*Manfrin et al., 2010*). Such findings are further supported by the analysis of the response of the Pacific oyster *Crassostrea gigas* to OA exposure using time-series (*Romero-Geraldo, Garcia-Lagunas & Hernandez-Saavedra, 2014*), showing a strong dependence on time and dose. Altogether, it seems that expression profiles can hardly be extrapolated to other conditions different to those being studied. Given the highly dynamic nature of the transcriptome, only consistent patterns in expression can be informative of environmental stress conditions (*Aardema & MacGregor, 2002*). This supports the use of expression signatures rather than individual biomarkers for biomonitoring purposes. Modeling systems of greater complexity including time and dose as variables would provide valuable information about the dynamics of the expression profiles.

The present work was completed by investigating the metabolic pathways associated with those enzymes identified as differentially expressed under OA exposure conditions (Supplemental Information 3). Most of these pathways are involved in energy production (e.g., glycolysis/gluconeogenesis pathway, the citrate cycle, the pentose phosphate pathway and the oxidative phosphorylation pathway) as well as the regulation of the cell cycle and metabolism of drugs and xenobiotics. The observed functional profiles are consistent between tissues and also with observations in other organisms and types of abiotic stress. Accordingly, the role of metabolic functions was observed at the proteomic level in the mussel *Perna viridis* exposed to OA pollution (*Huang et al., 2015*). The differential expression of enzymes involved in metabolic pathways such as Glycolisis, TCA and oxidative phosphorylation suggests that energy production becomes critical in situations of environmental stress. Such observations agree with the responses found in the Eastern oyster *Crassostrea virginica* exposed to different types of abiotic stress (*Chapman et al., 2011*). Furthermore, the role of the mTOR pathway as key regulator of the balance between energy consumption and cellular development was also evidenced in bivalves under environmental stress (*Clark et al., 2013*). An upregulation of enzymes PI3K, AMPK, LKB1 and ERK1/2 from this pathway (responsible for arresting the cell cycle when energy is required for resisting stress conditions) was found in the present work, suggesting that such mechanism is activated in the mussel as a response to OA toxicity. Lastly, the

differential expression of enzymes involved in immunity-related pathways like biosynthesis of antibiotics further supports a link between environmental stress and changes in the immunity system (*Malagoli et al., 2007*).

## CONCLUSIONS

The present work dissects the gene expression changes in different mussel tissues during early stages of DSP HAB episodes, suggesting that low concentrations of OA (below the legal OA limit established for safe consumption of shellfish) are enough to elicit sharp changes in the expression of genes involved in the response to this biotoxin. Prior to this work, a few studies attempted to investigate the transcriptomic changes in bivalves during HABs using high-throughput methods (*Manfrin et al., 2010*; *Suarez-Ulloa et al., 2013a*; *Gerdol et al., 2014*). However, the combined application of normalized and SSH libraries together with the development of a custom-made OA-specific microarray in the present work, provides a more comprehensive characterization of the OA-specific transcriptome, improving the accuracy of the analysis of expresion profiles compared to single-replicated RNA-seq methods (*Suarez-Ulloa et al., 2013a*). The custom-made microarray platform generated in this work represents a convenient tool for long-term monitoring projects, offering a good level of standardization with lower requirements in computational resources comparing to the otherwise more informative RNA-seq methodology (*Guo et al., 2013*). In addition, the transcriptomic coverage of this microarray is comparable to recent estimations for the size of the complete transcriptome in digestive gland of *Mytilus galloprovincialis* (*Gerdol et al., 2014*) as well as for the transcriptome of the Pacific oyster *Crassostrea gigas* (*Zhang et al., 2012*), thus representing a good approximation to an unbiased tool for expression analysis.

Our results suggest that the response to OA found in mussels is consistent with the model of intracellular response to stress previously reported for bivalve molluscs (*Anderson et al., 2015*). Accordingly, the activation of energy production mechanisms observed in the present work could be producing potentially harmful Reactive Oxygen Species (ROS), which unless controlled by chaperones or eliminated in the proteasomes, would induce apoptosis. An increase in ROS production has been recently reported for the mussels exposed to saxitoxins (i.e., neurotoxins responsible for the paralytic shellfish poisoning), supporting the applicability of this model to HABs exposure (*Astuya et al., 2015*). Indeed, our results show an upregulation in important chaperones (Hsp70) and proteases (cathepsins b and d) (Table 2) consistently with this model. Particularly the strong upregulation of cathepsins, known to be activated in the lysosomes (*Kagedal, Johansson & Ollinger, 2001*), in conjunction with the activation of transport mechanisms suggested by our results (Table 3), offer support to the lysosomal uptake hypothesis proposed by *Svensson, Sarngren & Forlin (2003)*. In addition, the upregulation of antimicrobial peptides suggests the activation of immunity mechanisms in conjunction with the general environmental stress response. However, it remains unclear whether this immune response is automatically triggered by abiotic factors or whether there is an opportunistic attack of

pathogens present in the microbiota of the mussels. Current efforts are directed to clarify this question (*De Rijcke et al., 2015*).

Further work studying more restricted conditions with shorter periods of exposure and lower concentrations of dinoflagellates would better inform about the sensitivity of the transcriptomic approach for the detection of OA-pollution in the ocean. Complementary, long-term monitoring projects in combination with meta-analysis of publicly available data could provide valuable information on the basal trancriptomic changes constituting a general environmental response as well as on the specific transcriptomic signature of DSP toxicity stress.

### Funding

The present work was supported by grants from the Biomolecular Sciences Institute (800005997) and start-up funds from the College of Arts and Sciences at Florida International University (JME-L), and from the Spanish Ministry of Economy and Competitivity (AGL2012-30897, JM). VS-U was supported by a Graduate Assistantship from the Department of Biological Sciences at FIU. VA-P was supported by the College of Engineering and Computing at Florida International University, under the supervision of Dr. Giri Narasimhan. The funders had no role in study design, data collection and analysis, decision to publish, or preparation of the manuscript.

### Grant Disclosures

The following grant information was disclosed by the authors:
Biomolecular Sciences Institute: 800005997.
College of Arts and Sciences at Florida International University.
Spanish Ministry of Economy and Competitivity: AGL2012-30897.

### Competing Interests

The authors declare there are no competing interests.

### Author Contributions

- Victoria Suarez-Ulloa conceived and designed the experiments, performed the experiments, analyzed the data, wrote the paper, prepared figures and/or tables, reviewed drafts of the paper.
- Juan Fernandez-Tajes conceived and designed the experiments, performed the experiments, analyzed the data, wrote the paper, reviewed drafts of the paper.
- Vanessa Aguiar-Pulido and M. Veronica Prego-Faraldo analyzed the data, reviewed drafts of the paper.
- Fernanda Florez-Barros performed the experiments, reviewed drafts of the paper.
- Alexia Sexto-Iglesias performed the experiments.
- Josefina Mendez conceived and designed the experiments, contributed reagents/materials/analysis tools, reviewed drafts of the paper.

- Jose M. Eirin-Lopez conceived and designed the experiments, analyzed the data, contributed reagents/materials/analysis tools, wrote the paper, prepared figures and/or tables, reviewed drafts of the paper.

## DNA Deposition

The following information was supplied regarding the deposition of DNA sequences:
   Normalized and SSH read datasets are available in the NCBI's Bioproject database under the accession number PRJNA167773.

## Data Availability

   The accession number for our raw microarray dataset in the GEO database is: GSE72817.

## Supplemental Information

Supplemental information for this article can be found online at http://dx.doi.org/10.7717/peerj.1429#supplemental-information.

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
