# Peer review of "Unbiased high-throughput characterization of mussel transcriptomic responses to sublethal concentrations of the biotoxin okadaic acid"

_PeerJ, doi:10.7717/peerj.1429_

## Round 0.1 · original submission · Minor Revisions

· Academic Editor

Minor Revisions

Please, consider all the suggestions in the revised manuscript.

·

Basic reporting

The manuscript is well written and organized. Figures and Tables are appropriate. The introduction is complete and exhaustive. No other major comment.

Experimental design

The overall experimental planning is scientifically sound and the appropriate methodologies have been used to perform the experimental challenges and bioinformatic analyses.
Just one minor comment: Line 180: “assuming a 100% amplification efficiency” is, in my opinion, an unnecessary generalization which in some specific cases could lead to significant errors in the calculation of expression values. The are means such as LinRegPCR to estimate amplification efficiency, which should be used to calculate corrected Cq values.

Validity of the findings

At least some of the differentially expressed genes in the digestive gland might not be linked to the presence of the toxin itself, but they are more likely to be connected to the different properties of the composition of P. lima vs T. suecica and I. galbana cells and cell walls. For example, this seems to be the case, in Table S2, of endo-beta xylanases and endo-beta glucanases. While the main conclusions of the authors remain valid, they should also take into consideration, with a brief statement in the discussion, the possibility that the gene expression profile observed is, at least to some extent, influenced by this factor. A “perfect” experimental design would have used a non-toxic strain of P. lima (if existing) in the control group instead of a continued feeding with T. suecica and I. galbana.

Lines 302-304: Other factors should be also taken into consideration: first of all, gene expression data cannot assess the effect of post-translational regulation (e.g. changes in protein turnover and activity). Second, even if the expression level of a certain gene appears to remain constant, the translation process could be negatively influenced by regulation by miRNA and other lncRNAs.

A very interesting point which is not remarked upon enough in the discussion in my opinion is the important downregulation of several genes associated with the transcriprion and cell division processes in the digestive gland (Table S4). Together with the upregulation of oxidative stress related enzymes, HSPs and apoptosis-related genes, the downregulation of "mRNA processing", "ribonucleoprotein complex biogenesis" and "transcription from RNA polymerase II promoter" really seem to suggest that digestive gland cells are struggling and therefore they drastically reduce transcription rates, maybe to save their energy budget. As a result, GO terms linked to cell division also seem to be down-regulated. Overall, this is consistent with the hypothesis pointing towards the involvement of mTOR as a major player in the response to OA.

Additional comments

A few specific comments concerning the immune-related transcripts:
-was the partial mytimacin sequence identified mytimacin-5? It would be interesting to clarify this point as MM5 is unique compared to the other mytimacins due to the presence of an additional disulphide bridge and C-terminal extension, whose functional significance is unknown.
-The transcript listed in table S2 as mytilin-1 is really pertaining to the mytilin AMP family? Unfortunately there are two sequences named “mytilin-1” in public databases whose annotation is misleading (P86853.1 and AKI87978.1), as they have nothing to do with the AMP family extensively studied over the past two decades. The authors should check this annotation and modify the text if the annotation was derived from one of the two accessions listed above.
-Overall, the authors find several upregulated transcripts linked to innate immunity in the digestive gland. Among these, some have been tightly linked to a specific expression in hemocytes (mytilin,mytimycin and MgC1q3), which could be an indication of infiltration of hemocytes in the digestive gland tissue, a response which has been also observed in mussels in response to other unrelated algal toxins (PSTs) (see Galimany et al., 2008). In addition, somewhat similar results, in terms of gene expression, have been observed in mussels exposed to PSP, with the upregulation of oxidative stress related genes, HSPs and mytilin B (see Nunez-Acuna et al. 2013 for Mytilus chilensis). Overall, this information further support one of the main conclusions of this work, which is “the activation of immunity mechanisms in conjunction with the general environmental stress response.” or even, more specifically, with toxin accumulation in the digestive gland.
-Line 320-321: To improve this sentence, instead of “Although C1q…” I would suggest to use “Although no clear homologs to the vertebrate C1q complex subunits have not been found in invertebrates yet…”. Furthermore, I would suggest to replace the following sentence with “the C1q domain-containing protein family underwent massive expansion in bivalves, including Mytilus spp” and the most appropriate reference here would be Gerdol et al. 2015 (DCI) instead of the 2011 reference.

In the caption of Tables S4 and S5 “enriched for the set of upregulated transcripts” should be replaced by “enriched for the set of upregulated and downregulated transcripts”. Furthermore, indicating the p-value for enrichment and the number of expected and observed transcripts would be beneficial.

Reviewer 2 ·

Basic reporting

The authors used a genome-wide approach to investigate transcriptomic responses of mussels to early bloom OA concentrations, which is an unexplored and important line of study. The authors show that there are significant transcriptomic responses at low OA levels and add to the known molecular responses to OA exposure, which is relevant to the field. Overall, the background and rationale for the study and approach are well reasoned and described clearly.

The main critique of the manuscript is the presentation of the data. The details of the transcriptomic responses are in supplemental material while some of the figures in the main text are not very informative. Figure 1, which shows the redundancy between the cDNA libraries, is one such example. The figure shows that there is redundancy between libraries and provides the rationale for combining the libraries to make the microarray probes, but the authors never discuss comparisons between the different libraries in detail. Because it is not a key component of the discussion of the main focus of the paper, this could be moved to supplemental. Similarly, Figure 5, because it represents broad GO categories, is too general to be informative.

Instead, the authors should move Supplemental Material S2, S3, S4, and S5 to the main text. The authors describe specific transcripts that are differentially regulated, and that data should be in the main text. The list of differentially regulated transcripts and GO annotations are not so large that it would overwhelm the text. I would also recommend grouping the transcripts by functional category in the tables to make it easy to see which transcripts are differentially expressed in the larger categories discussed by the authors (ie. proteosomal activity, molecular transport, cell cycle regulation, etc.). The authors could also decide to only show the transcripts they specifically comment on (HSPs, cathepsins, mytimacin, etc.) and include the full list in Supplemental. Because the lists are not very long, this probably would not be necessary. If the supplemental files are brought into the main text, their legends should be clarified to indicate the lists of transcripts come from the microarray data. Additionally, the authors discuss upregulation of cAMP-responsive element binding protein (CREB), vdg3, and elongation factor 2 in the context of corroborating previous studies, but the authors should explain their significance in the context of their study, as well. How do these transcripts fit in with the other differentially regulate transcripts? Further, CREB is not on the lists provided in the supplemental material.

It is not clear in Figure 5, S4, S5, and the discussion of “Expression of function profiles of transcripts differentially expressed in response to OA” (page 18) if the data presented from GO annotation is from the microarray data or the cDNA libraries. Based on its juxtaposition in the text, it is assumed it is from the microarray data, but the methods discuss similar analysis of the cDNA libraries (page 9). It should be stated in the text and figure legends that the GO annotation presented is from data from the microarray, if indeed that is accurate. Further, the methods describe mapping the transcripts to the KEGG database, which seems to be discussed on page 20 and in S6, but this also needs to be clearly stated.

There are a few typos in the text that need attention. On page 5, line 67, a noun is missing (“OA is the main responsible of acute DSP…”). This line is repeated in the abstract and should be corrected in both places. Page 18 line 334 states “the overall top ten of most represented…” when the tables show the top twenty GO terms. Page 19, line 354 correct “findings.”

Experimental design

The research question is clearly stated and the overall approach is appropriate. A problem in the experimental design is the qPCR data. In this study, qPCR was used to validate transcripts observed in the cDNA libraries but not in the microarray studies, which is typically standard. Only some of the transcripts analyzed by qPCR, HSP70 and GTPases, are differentially regulated in the microarray studies while the other transcripts are not found in the supplemental files of the array data and are not discussed further. If the authors were aiming to validate the cDNA libraries, this data is fine to include, but the authors also need to validate the microarray data as they discuss several specific transcripts that are differentially regulated based on the microarray data. Further, the authors assume 100% amplification efficiency (Page 11 line 180). When using SYBR green for qPCR, standard curves should be generated for each set of primers used to show amplification efficiency.

Validity of the findings

The authors’ analyses of the data are appropriate, and their conclusions are supported by the data on the condition that the microarray data is validated by a second method (ie. qPCR). Given that previous studies have shown tissue-specific effects, the discussion of this aspect of the data comparing the gill and digestive gland could be expanded upon. While in general there are a lot of similarities in the functional categories, which the authors state, there are also some distinctions between the groups. Also, it is possible the low number of differentially expressed transcripts in the gill is the result of the microarray being developed from the digestive gland cDNA libraries or the result of a lower toxin load as stated in the methods, which could also be stated in the discussion. This study provides additional evidence in understanding the response of mussels to OA at the molecular level, provides leads for biomarker development, and developed a microarray that can become a valuable tool for both basic and applied research in the field.

·

Basic reporting

The structure of the article is clear, and the tables and figures are relevant and illustrate nicely the concepts developed in the text.

Experimental design

The question posed by the authors is well defined, and the methods are sound, appropriate and well described.

Validity of the findings

The analysis is statistically sound, and the authors make their data available to promote its inclusion in future meta-analysis.

Additional comments

In this manuscript, Suarez-Ulloa et al. characterize the transcriptomic response to low concentrations of okadaic acid. The monitoring of this pollutant is of importance for human health and it has commercial applications since fisheries are periodically closed due to the high levels of toxins derived from algae blooms. Previous studies in molluscs have focused on high concentrations of the pollutant. Therefore, this work is a nice addition to its field because it investigates systematically the transcriptomic effects of the pollutant bellow the limit established by law for safe consumption of shellfish. I would like to recommend the manuscript for publication after addressing the following comments.

* Perhaps my major criticism is that discussion could be improved for clarity. The authors declare in the introduction that “additional efforts are still required to transform the extraordinary amount of molecular data resulting from omic experiments into sensible and rapid biomarkers of marine pollution” (line 89) and also the relevance of this study “for developing molecular biomarkers of marine pollution during DSP blooms” (line 109). It is obvious that this study has generated additional data but authors should clarify how this data could translate into measurements more useful than the OA quantification using high resolution mass spectrometry, for example, is the transcriptomic signature transferable to other pollutants?

* Suarez-Ulloa et al. suggest in conclusions that further work should be directed towards generating more data with shorter and longer periods of exposure (line 423). I think this avenue for research is very promising. This study is already a coherent body of work but in my opinion further work should be aimed to perform qPCRs of the most promising markers in a number of individuals non included in the original pool. And, second, to build a model with the results obtained in these individuals. Initially this model could be logistic (expose/non-exposed) and would give weight to each of the transcripts, and when more doses are tested and time is included as one of the variables it would be possible to build a more complex but useful model. I think that modelling should be mentioned more explicitly in discussion after authors state that “expression profiles can hardly be extrapolated to other conditions different to those being studied” (line 358) to encourage the inclusion of this data in future meta-analysis. I think it is unclear in this context what authors refer to as “time-resolved” vs “qualitative assessment” later in the same paragraph.

Minor points:
* Please describe the samples more thoroughly since transcriptomic data can be dependent on age and reproductive cycle stage.
* line 321. The genus name should be in italics.
* The literature search has been performed properly but the reference list contains some typos:
Title is in capital letters. lines 460 484 487 559 567 584 599.
The first name of one of the authors is spelled, line 484 (also in the text, line 104).
The names of the species should be in italics. lines 461 478 504 551 561 569.

---

## Round 0.2 · accepted · Accept

· Academic Editor

Accept

After your revision the paper has been deeply improved.